# Lesion Conspicuity in Contrast-Enhanced Mammography: A Retrospective Analysis of Tumor Characteristics

**DOI:** 10.3390/cancers17030501

**Published:** 2025-02-03

**Authors:** Chiara Bellini, Tommaso Susini, Kassandra Toncelli, Martina Pandolfi, Giuliano Migliaro, Francesca Pugliese, Bianca Vanzi, Ludovica Incardona, Giulia Bicchierai, Federica di Naro, Diego de Benedetto, Sofia Vidali, Silvia Pancani, Vittorio Miele, Jacopo Nori Cucchiari

**Affiliations:** 1Breast Imaging Unit, Department of Radiology, Azienda Ospedaliero, Universitaria Careggi, 50134 Firenze, Italy; kassandra.toncelli@unifi.it (K.T.); martina.pandolfi@unifi.it (M.P.); giuliano.migliaro@unifi.it (G.M.); francescesca.pugliese@unifi.it (F.P.); bianca.vanzi@studio.unibo.it (B.V.); incardonaludovica@gmail.com (L.I.); bicchieraig@aou-careggi.toscana.it (G.B.); dinarof@aou-careggi.toscana.it (F.d.N.); debenedettod@aou-careggi.toscana.it (D.d.B.); vidalis@aou-careggi.toscana.it (S.V.); norij@aou-careggi.toscana.it (J.N.C.); 2Breast Unit, Gynaecology Section, Department of Health Sciences, University of Florence, 50121 Firenze, Italy; tommaso.susini@unifi.it; 3IRCCS Fondazione Don Carlo Gnocchi Onlus, 50143 Firenze, Italy; silvia_pancani@hotmail.it; 4Department of Experimental and Clinical Biomedical Sciences, University of Florence, 50121 Firenze, Italy; vmiele@sirm.org; 5Department of Radiology, Azienda Ospedaliero, Universitaria Careggi, 50134 Firenze, Italy

**Keywords:** contrast-enhanced mammography, breast cancer, lesion conspicuity

## Abstract

Contrast-enhanced mammography (CEM) is a functional imaging technique that utilizes the intravenous injection of iodinated contrast media and is increasingly used in the diagnosis of breast cancer. The term “lesion conspicuity” (degree of enhancement in comparison to the background) was introduced in the latest CEM lexicon released by the American College of Radiology (ACR). However, the ACR lexicon does not establish a clear association between high conspicuity and the likelihood of malignancy, making it crucial to understand the factors that influence conspicuity. This study aims to explore how tumor characteristics such as size, enhancement type, and molecular subtypes impact lesion conspicuity in CEM. By identifying these factors, the findings can help radiologists interpret CEM results more accurately, improving the detection and evaluation of breast cancer.

## 1. Introduction

Breast cancer (BC) is the most common cancer in women, and although mortality rates are slightly decreasing, it remains one of the leading causes of death due to oncological pathology [1]. Consequently, optimizing early diagnostic methods and accurate locoregional staging is essential to avoid the risks of both under- and over-treatment in patients requiring surgery.

In this context, contrast-enhanced mammography (CEM) has emerged as a promising tool in breast imaging. Thanks to the intravenous contrast agents used in CEM, we can detect areas of increased and irregular blood flow (neo-angiogenesis) which are often associated with malignancies [2].

There is evidence that CEM is indicated in determining the true extent of disease especially in a presurgical assessment, in evaluating responses to neoadjuvant chemotherapy, in problem solving and in screening or follow-up of patients at high risk of developing breast cancer [3]. CEM has also become widely used in clinical practice due to its low costs, its better tolerability by patients and its greater accessibility compared to MRI [4].

The role of contrast-enhanced mammography in the specific setting of locoregional staging of breast cancer is well established [2,5]. It is particularly valuable due to its high sensitivity in identifying the index lesion, detecting additional occult lesions that may not be visible on conventional imaging, and providing accurate size assessment of the tumor [6,7,8]. This detailed evaluation can significantly influence treatment planning, leading to a change in the surgical approach in approximately 20% of cases [9]. Its utility in ensuring more precise surgical and therapeutic decisions makes it a crucial tool in the preoperative setting.

Due to the relatively recent adoption of CEM, a standardized lexicon for its reporting was initially lacking, and until recently, radiologists relied on MRI descriptors [10,11]. To address this gap and improve the diagnostic performance of this new radiological technique, the American College of Radiology (ACR) published a dedicated supplement in 2022 [12].

Among the descriptors of the enhancing findings, in addition to morphology, margins and internal pattern of enhancement, a new parameter called “lesion conspicuity” was introduced. Lesion conspicuity (LC) is defined as the degree of lesion enhancement relative to the background parenchymal enhancement (BPE) and can be classified as low, moderate or high, but these classifications however remain largely subjective and qualitative. Low conspicuity means that the enhancement is equal to or slightly greater than BPE, while high conspicuity means that the enhancement is much greater than BPE, and moderate that the enhancement is between low and high. Currently, the recognition and management of specific levels of lesion conspicuity remain a topic of debate, and the ACR itself highlights the lack of data correlating lesion conspicuity with malignancy risk, so the terms for conspicuity are included in the lexicon to encourage future research.

In breast MRI, the level of enhancement is useful to distinguish between benign and malignant lesions, contrast uptake of malignancy being earlier and more intense due to tumoral neo-angiogenesis [13]. It is also known that higher conspicuity levels often correlate with aggressive tumor features compared to “less aggressive” cancers [14].

Given the limited literature that clearly assesses the impact of lesion conspicuity on clinical practice [15,16], the aim of our study was to evaluate the impact of tumor characteristics on lesion conspicuity in contrast-enhanced mammography (CEM) and specifically to identify factors associated with different levels of conspicuity, such as the histological–biological profile or imaging features such as the type of enhancement and the actual lesion size.

## 2. Materials and Methods

### 2.1. Patients’ Population

This study, conducted under the Declaration of Helsinki, was approved by the Institutional Review Board (IRB) of the Azienda Ospedaliero-Universitaria Careggi (#16.251_AOUC). Additional informed consent was waived by the IRB due to the retrospective nature of the study.

In this single-center observational retrospective study, we reviewed consecutive cases of patients undergoing CEM as presurgical locoregional staging for breast cancer at our Breast Imaging Unit from January 2018 to December 2021. CEM is routinely performed in locoregional presurgical staging on all patients in our unit, regardless of breast density, histotype, or tumor size. CEM is not used preferentially in dense breasts; rather, it is routinely conducted before surgery in our center, so this approach has contributed to minimize bias in the retrospective collection of data.

Inclusion criteria were histologic diagnosis of breast cancer on core-needle biopsy (CNB) or vacuum-assisted biopsy (VAB), execution of CEM as preoperative locoregional staging, breast surgery performed in our center, with availability of complete histologic reports of CNB, VAB and surgical specimens.

Exclusion criteria were patients who received neoadjuvant chemotherapy, which could have made the final results less reliable; patients with multifocal or multicentric disease and patients with index lesions that showed no enhancement on CEM; patients who referred to other institutions and with contraindications to CEM.

Multifocal or multicentric diseases were excluded due to the complexity they introduce in the analysis of lesion conspicuity. Multifocal and multicentric tumors are often heterogeneous, which complicates the assessment of conspicuity. To ensure a more coherent evaluation, these types of lesions were excluded from the study.

### 2.2. Histology

Stereotactic-guided VAB was performed using a vacuum-assisted biopsy device (Mammotome revolve; Devicor Medical Products, Cincinnati, OH, USA), with an 8-gauge needle, with the patient lying on a digital prone table (Affirm Prone Biopsy System; Hologic, Marlborough, MA, USA). An average of 12 core samples per lesion were obtained.

Percutaneous CNB was performed with a semi-automated biopsy gun (Precisa, Hospital Service, Rome, Italy) with a 14 G, 10 cm-long needle. A mean of three core samples per lesion was obtained.

For each lesion, the biopsy results and the final histology after surgery with the ER, PgR, C-erb-2 status and Ki-67 index were analyzed and collected by two pathologists with more than 25 years of experience in breast pathology.

### 2.3. CEM Protocol

CEM was obtained by using a Selenia Dimensions mammography system (Hologic) able to perform full-field 2D digital mammography, 3D tomosynthesis and the proper examination with the contrast medium.

A total volume of 1.5 mL/kg of body weight low-osmolarity iodine-based contrast material (Iopamiro 370, Bayer HealthCare LLC, Whippany, NJ, USA) was intravenously administered with an automated bolus injection having a flow rate of 3 mL/s and followed by a 20 mL saline flush.

Scanning started from 2 min after the beginning of injection, and every mammographic view (the standard CC and MLO for each breast) was obtained.

CEM was unrolled in quick succession using different values of voltage during acquisition for each main view of the breast.

Low-energy (LE) images were developed with a tube voltage of 26–31 kVp and a rhodium or silver filter. In this way, they were comparable to the morphological images obtained by conventional mammography.

High-energy (HE) images are acquired with a voltage of 45–49 kVp and a copper filter. These images are not suitable for diagnostic purpose because of their complete opacity.

A recombination algorithm, throughout a subtraction between the high and low energy images, provided a final image in which only the areas of contrast enhancement were highlighted, defined as recombined images (RCs) [17].

### 2.4. Image Analysis

All CEM images acquired in the four standard projections were reviewed by three experienced breast radiologists with 4–27 years of experience in breast imaging. In cases where there was no agreement among the radiologists regarding lesion conspicuity, the final decision was made through consensus.

In the recombined images, the three radiologists assessed the following features: the level of BPE of normal glandular tissue in accordance with the BI-RADS criteria, classifying it as minimal, mild, moderate, or marked; the size of the index lesion with detectable suspicious enhancement measured in millimeters (mm); the type of enhancement, mass, non-mass (NME), enhancing asymmetry (EA), or a combination of these when a lesion showed a mass enhancement next to a non-mass enhancement; and, most importantly, the lesion conspicuity (LC), which was defined as low (1), moderate (2), and high (3) after a qualitative visual assessment, as described from the ACR lexicon supplement for CEM [12]. Conspicuity was defined as low if the enhancement of the lesion was equal to or slightly greater than BPE, high if the enhancement was much greater than BPE, and moderate if the enhancement was between low and high, according to its definition in the ACR lexicon supplement [12].

LC was visually assessed in a qualitative way, and to ensure consistency in determining LC, the radiologists jointly reviewed imaging from 30 patients as a case test (excluded from the study analysis), with 10 low LC, 10 moderate LC, and 10 high LC.

We also reviewed the low-energy images in particular to accurately define the extent of a lesion when it showed low intensity in the recombined images.

We excluded any lesion that showed the enhancement pattern defined as “rim enhancement”, in which only the lesional margins are highlighted by the contrast agent. This choice was made because, in our series, this was a finding often associated with biopsy outcomes, which consequently no longer allowed an adequate measurement of the actual lesion size. In the few cases where it was not related to biopsy, it remained a feature prone to subjective interpretation. We also did not include two patients whose lesions were outside the field of view and five patients with no lesional enhancement (they all had a diagnosis of pure DCIS) who therefore could not be described further.

### 2.5. Statistical Analysis

Statistical analysis was performed using SPSS IBM version 28.0 (Armonk, NY, USA: IBM Corp). Continuous variables were checked for normality using the Shapiro–Wilk test. Non-normal continuous variables were expressed as median and interquartile range. Categorical and dichotomous variables were expressed as frequency and percentage. Distribution of clinical and pathological features of breast cancer was compared among different conspicuity groups (low = 1, moderate = 2, and high = 3) using the Chi-square test for dichotomous and categorical variables, and the Jonckheere–Terpstra test for continuous variables. Post hoc paired comparisons were conducted using the Mann–Whitney U test in the latter case. In addition, post hoc power analysis was performed for the variables that were found to be statistically significant using the G*Power software v 3.1.9.4. Those variables that resulted as significantly different in distribution according to lesion conspicuity values were included as independent variables in an ordinal logistic regression model with lesion conspicuity as the dependent variable. The α-level was fixed at 0.05 for all statistical tests except the Mann–Whitney U tests after the Jonckheere–Terpstra test, where *p* was reduced according to Bonferroni correction.

## 3. Results

Finally, 552 CEMs were selected with a median patient age of 62.3 years (range 35–90), for a total of 552 biopsy-proven malignant lesions.

Regarding enhancement types observed on CEM, the majority of lesions appeared as masses (78.1%), followed by NME (16.8%), mass + NME (4.0%), and EA (1.1%). The molecular subtype distribution was as follows: 44.5% of cases Luminal A, 31.4% Luminal B, 19.9% HER2-positive, and 4.2% triple-negative breast cancer (TNBC). In terms of final histology, invasive ductal carcinoma was the most prevalent (56.3%) with associated DCIS observed in 22% of cases.

Descriptive variables of the study population are described in detail in Table 1.

On the basis of lesion conspicuity, the 552 cases were classified as follows: low conspicuity (1) = 109 (19%); moderate conspicuity (2) = 276 (50%); high conspicuity (3) = 171 (31%).

An example of lesions with low, moderate, and high conspicuity is shown in Figure 1.

The correlation between lesion conspicuity and the different clinical pathologic tumor features is shown in Table 2.

Significant associations were observed between lesion conspicuity and enhancement type on CEM (*p* < 0.001), lesion dimensions (*p* < 0.001), Ki67 index (*p* = 0.019), molecular subtype (*p* = 0.025), and final histology after surgery (*p* = 0.001). Specifically, conspicuous lesions (higher conspicuity scores) were more likely to present as masses or mass + NME, with increasing dimensions and Ki67 levels also associated with higher conspicuity. Luminal A subtype was more common in lesions with lower conspicuity, whereas HER2-positive was associated with higher conspicuity scores. The power analysis performed for those variables that showed significant difference yielded a 99% power for “Enhancement type on CEM” and “Dimensions”, 71% for “Ki67 index”, 83% for “Molecular subtype”, and 97% for “Histology”.

The ordinal logistic regression model, showed in Table 3, performed to control for confounding factors and identify independent predictors of lesion conspicuity, showed that lesion dimensions and enhancement type on CEM were significant independent predictors of lesion conspicuity.

Larger lesion dimensions were significantly associated with higher conspicuity (*p* < 0.001). Additionally, masses (*p* < 0.001), Mass + NME (*p* = 0.003), and EA (*p* < 0.001) were associated with higher conspicuity compared to NME, which served as the reference category.

## 4. Discussion

In recent years, CEM has emerged as an important diagnostic tool in the diagnosis and preoperative work-up of breast cancer. As the experience in its use increases, the lexicon utilized in radiological reports also needs to be refined and standardized. The purpose of our retrospective study was to determine the relationship between lesion conspicuity, a new descriptor of contrast-enhanced mammography in the ACR lexicon and specific clinical, radiological, and pathological features of breast cancer.

To our knowledge, limited research is available on this topic. A previous study analyzed the correlation of lesion conspicuity on CEM with lesion type on mammography (MG), and histological results concluded that findings with strong or medium enhancement were more frequently associated with malignancy and mass-like lesions on MG [18].

A study by Rudnicki et al. compared the level of enhancement in CEM and MRI kinetics curves and found an association between these two factors: lesions with wash-out curve on MRI more often presented strong enhancement on CEM, while in lesions with a progressive enhancement curve, strong enhancement on CEM was the rarest [19]. Another study showed that the combination of LC analysis and kinetics assessed in CEM between early and delayed (after 8 min) RC images could enhance CEM diagnostic accuracy [15].

Another study by Li et al. investigated the correlation between CEM imaging characteristics and different molecular subtypes of breast cancer and found no association between lesion conspicuity and molecular subtypes [20], in contrast with the study of Nicosia et al., where a significant association of LC with receptor status was reported [16]. Another study by Marzogi et al. confirmed the association between LC and breast cancer aggressivity hallmarks, even though the author warned that the absence of enhancement should not be used to downgrade suspicious calcifications [21].

Some studies have attempted to perform a quantitative analysis of lesion conspicuity using semi-automatic segmentation programs, underscoring the potential ability to distinguish between benign and malignant lesions by this approach [22,23,24].

Our results showed a significant association of lesion conspicuity with certain cancer features, most notably lesion size, type of enhancement, histology and molecular subtype, although only lesion size and type of enhancement remained significant and independent in multivariate analysis.

Regarding histologic subtypes, invasive ductal carcinoma (IDC) lesions were more conspicuous than ductal carcinoma in situ (DCIS); this disparity may be due to the limited vascularity in DCIS, as well as the more dispersed growth patterns typical of non-invasive tumors, which can limit contrast uptake and reduce imaging conspicuity. This result supports previous studies that indicate that DCIS may present absent enhancement or low conspicuity in CEM [15,21,25], making the detection of DCIS challenging, but CEM sensitivity and specificity can be enhanced by the evaluation of the low-energy (LE) images [26], which are comparable to standard mammography in terms of quality [27,28]. Additionally, we noted that lobular carcinoma tends to distribute more uniformly across various LC degree groups.

In our analysis, features of cancer aggressiveness, such as levels of Ki-67 ≥ 20%, and the HER2-positive molecular subtype, were associated with higher conspicuity in univariate analysis; in contrast, Luminal A tumors were more commonly observed in the low-LC group.

This result is in line with the previous literature, in particular with the study of Nicosia et al. who reported a significant correlation between high lesion conspicuity and biomarkers of breast cancer aggressiveness like absence of expression of ER/PgR, levels of Ki-67 ≥ 20% and G3 tumors [16].

The significant correlation between high conspicuity and aggressive molecular subtypes, although it was not an independent predictor in the current study, provides a basis for considering CEM’s role in stratifying patients by risk. Hence, the finding of a lesion with high conspicuity on CEM may prompt more invasive and faster diagnostic procedures.

We found a statistically significant association between type of enhancement and level of conspicuity. Indeed, our results showed that mass-type enhancement was more common in lesions with higher conspicuity, while non-mass enhancement was frequent in lesions with low conspicuity. We did not find previous studies addressing this specific topic on CEM; however, this result appears logical, given the three-dimensional nature of masses which could lead to higher contrast uptake. In our series, EA showed the stronger association with high conspicuity, not only more than NME, but even more than mass enhancement. However, given the limited sample size of EA of only six cases in our series, this finding should be interpreted with caution.

In our study, we found that higher levels of lesion conspicuity in CEM were also strongly correlated with lesion size by multivariate analysis. In our opinion, the significant association between high LC and larger size in CEM may be partially attributed to the two-dimensional nature of this imaging technique. Unlike three-dimensional imaging modalities, which provide volumetric assessment, CEM captures a flat, two-dimensional representation of the breast. This may lead to enhanced conspicuity in larger lesions simply because they occupy more visual space within the image.

Our results suggest that while higher conspicuity is often associated with malignancy, it is not exclusive to malignant lesions; benign findings may have high LC depending on factors like size and vascular characteristics. This points to an important consideration: a high conspicuity score should prompt further diagnostic evaluation but not be regarded as definitive of malignancy on its own.

Additionally, our findings underscore the importance of not dismissing masses with low conspicuity as benign without thorough assessment, particularly in the presurgical setting since some cancers may present with only subtle enhancement due to factors like low vascularity or smaller lesion size. This highlights a critical point for clinical practice: both highly conspicuous and faintly enhancing lesions warrant careful evaluation, as relying solely on conspicuity as a marker for malignancy could lead to under-diagnosis of less obvious cancers. In the context of presurgical planning, where accurate staging and detections of all malignant lesions are essential, a cautious approach toward low-conspicuity lesions is recommended.

We acknowledge that this study has some limitations. First of all, this is a single-center series. Indeed, this setting may affect the generalizability of our findings, as CEM equipment and protocol may vary across institutions. Moreover, the subjective qualitative grading of conspicuity remains a challenge, as interpretation can vary between radiologists, especially considering that to date, software for quantitative enhancement assessment in CEM is not widely diffused. A standardized grading system could further improve consistency and diagnostic reliability. The 2022 ACR lexicon supplement for CEM has laid the groundwork for such standardization, but further refinement is needed.

Additionally, another limitation is that our study population was entirely composed of women with histologically confirmed malignant lesions; this selection process precludes the assessment of CEM’s effectiveness in differentiating between benign and malignant findings, an important area for future study. Future studies should consider larger, multicenter cohorts of patients and include benign lesions to evaluate CEM’s specificity and sensitivity more comprehensively. Finally, the variable “Ki67 index” showed a power slightly below 80%, suggesting that caution should be used when interpreting the results obtained and that further studies involving larger sample sizes should be performed.

Future research should focus on validating these findings in a multicentric fashion and examining whether quantitative methods of measure and the development of machine learning algorithms can provide more objective and reproducible assessments. By confirming the link between lesion size, conspicuity, and malignancy, and incorporating these parameters into standardized diagnostic protocols, CEM could become an even more valuable tool for identifying high-risk lesions, potentially optimizing treatment pathways and outcomes for patients.

## 5. Conclusions

In conclusion, our study suggests that CEM conspicuity is significantly influenced by lesion size and enhancement type, with larger and mass-type lesions being more conspicuous. The associations with molecular subtype and proliferative index hint at CEM’s potential utility in identifying aggressive cancers, although further research is needed to confirm these findings.

## Figures and Tables

**Figure 1 cancers-17-00501-f001:**
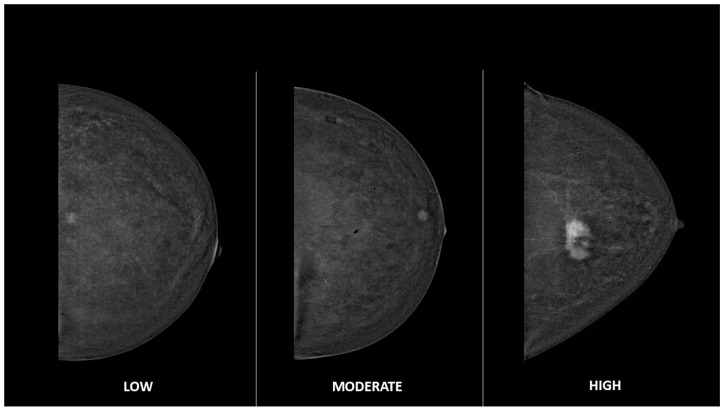
Example of lesions with low, moderate, and high conspicuity.

**Table 1 cancers-17-00501-t001:** Characteristics of the study sample.

Variable	*n* (%), [IQR ^1^]
Age (years, median)	62.3 [20.4]
Enhancement type on CEM	
EA	6 (1.1)
Mass	431 (78.1)
Mass + NME	22 (4.0)
NME	93 (16.8)
Dimensions (mm, median)	17 [13]
ER (positive)	515 (93.3)
PgR (positive)	489 (88.6)
HER2 (positive)	100 (19.9)
Grade of differentiation	
G1	122 (22.1)
G2	260 (47.1)
G3	170 (30.8)
Molecular subtype	
Luminal A	224 (44.5)
Luminal B	158 (31.4)
HER2-positive	100 (19.9)
TNBC	21 (4.2)
Histology	
Mixed ductal and lobular carcinoma	39 (7.1)
Invasive ductal carcinoma	311 (56.3)
Invasive lobular carcinoma	66 (12.0)
DCIS	46 (8.3)
Other	90 (16.3)

^1^ IQR: interquartile range.

**Table 2 cancers-17-00501-t002:** Distribution of clinical and pathological features of breast cancer among different lesion conspicuity groups.

Variables	Lesion Conspicuity			*p*-Value
	1 *n* = 105	2 *n* = 276	3 *n* = 171	
	*n* (%) [IQR]	*n* (%) [IQR]	*n* (%) [IQR]	
Enhancement type on CEM				<0.001
EA	0 (-)	1 (0.4)	5 (2.9)	
Mass	70 (66.7)	216 (78.3)	145 (84.8)	
Mass + NME	1 (1.0)	13 (4.7)	8 (4.7)	
NME	34 (32.4)	46 (16.7)	13 (7.6)	
Dimensions (mm, median)	12 [8]	16 [15]	20 [15]	<0.001
ER (positive)	96 (91.4)	262 (94.9)	157 (91.8)	0.307
PgR (positive)	92 (87.6)	249 (90.2)	148 (86.5)	0.467
HER2 (positive)	12 (13.5)	47 (19.2)	41 (24.3)	0.111
Grade of differentiation				0.770
G1	26 (24.8)	61 (22.1)	35 (20.5)	
G2	51 (48.6)	131 (47.5)	78 (45.6)	
G3	28 (26.7)	84 (30.4)	58 (33.9)	
Ki67 index (%)				0.019
<20%	36 (40.4)	75 (30.6)	40 (23.7)	
≥20%	53 (59.6)	170 (69.4)	129 (76.3)	
Molecular subtype				0.025
HER2-positive	12 (13.5)	47 (19.2)	41 (24.3)	
Luminal A	50 (56.2)	112 (45.7)	62 (36.7)	
Luminal B	21 (23.6)	80 (32.7)	57 (33.7)	
TNBC	6 (6.7)	6 (2.4)	9 (5.3)	
Histology				0.001
Mixed ductal and lobular carcinoma	5 (4.8)	16 (5.8)	18 (10.5)	
Invasive ductal carcinoma	53 (50.5)	148 (53.6)	110 (64.3)	
Invasive lobular carcinoma	12 (11.4)	35 (12.7)	19 (11.1)	
DCIS	14 (13.3)	30 (10.9)	2 (1.2)	
Other	21 (20.0)	47 (17.0)	22 (12.9)	

IQR: interquartile range.

**Table 3 cancers-17-00501-t003:** Ordinal logistic regression model.

Variable	β	Std. Error	Wald	df	Sig.	95% Confidence Interval	
						Lower Bound	Upper Bound
Dimensions (mm)	0.064	0.008	58.637	1	0.000	0.048	0.081
Enhancement type on CEM							
EA	4.819	1.192	16.344	1	0.000	2.482	7.155
Mass	2.234	0.336	44.141	1	0.000	1.575	2.893
Mass + NME	1.541	0.512	9.049	1	0.003	0.537	2.544
NME = ref	-	-	-	-	-	-	-
Ki67 index (%)							
<20%	−0.172	0.0251	0.473	1	0.492	−0.664	0.319
≥20% = ref	-	-	-	-	-	-	-
Histology							
Mixed ductal and lobular carcinoma	0.325	0.349	0.866	1	0.352	−0.359	1.009
Invasive lobular carcinoma	−0.124	0.286	0.188	1	0.665	−0.685	0.437
Other	−0.175	0.252	0.480	1	0.488	−0.669	0.320
DCIS	−0.585	1.959	0.089	1	0.765	−4.424	3.254
Invasive ductal carcinoma = ref	-	-	-	-	-	-	-
Molecular subtype							
Luminal B	0.059	0.253	0.055	1	0.815	−0.436	0.555
HER2-positive	0.417	0.266	2.465	1	0.116	−0.104	0.938
TNBC	−0.418	0.474	0.777	1	0.378	−1.348	0.512
Luminal A = ref	-	-	-	-	-	-	-

Ref = reference category; EA = enhancing asymmetry; NME = non-mass enhancement; DCIS = ductal carcinoma in situ; TNBC = triple-negative breast cancer.

## Data Availability

The data presented in this study are available on request from the corresponding author due to privacy restrictions.

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
