# Peer review of "Lesion Conspicuity in Contrast-Enhanced Mammography: A Retrospective Analysis of Tumor Characteristics"

_cancers, 2025, doi:10.3390/cancers17030501_

Round 1
Reviewer 1 Report
Comments and Suggestions for Authors
Recommendation: Major review
In this manuscript, the author reports, "Lesion Conspicuity in Contrast-Enhanced Mammography: A Retrospective Analysis of Tumour Characteristics". The authors should address the following questions before getting publication.
1. How was the sample size of 552 patients determined? Was a power analysis conducted to ensure adequate statistical power?
2. clarify how lesion conspicuity was quantitatively measured or was it solely based on qualitative assessment?
3. What specific criteria did the radiologists use to categorize lesion conspicuity into low, moderate, and high levels?
4. why multifocal or multicentric diseases were excluded, given their common occurrence in breast cancer?
5. How did the study account for inter-observer variability among the radiologists assessing the lesion conspicuity?
6. What measures were taken to minimize bias in the retrospective collection of data?
7. justify the use of univariate and multivariate analyses in their study? Could other statistical methods have been more appropriate?
8. provide more details on the statistical significance of the histological subtypes and their impact on lesion conspicuity?
9. What are the clinical implications of the findings for radiologists using CEM in routine practice?
10. What are the potential biases in the study due to its retrospective design, and how might these have been mitigated?
Comments on the Quality of English Language
Recommendation: Major review
In this manuscript, the author reports, "Lesion Conspicuity in Contrast-Enhanced Mammography: A Retrospective Analysis of Tumour Characteristics". The authors should address the following questions before getting publication.
1. How was the sample size of 552 patients determined? Was a power analysis conducted to ensure adequate statistical power?
2. clarify how lesion conspicuity was quantitatively measured or was it solely based on qualitative assessment?
3. What specific criteria did the radiologists use to categorize lesion conspicuity into low, moderate, and high levels?
4. why multifocal or multicentric diseases were excluded, given their common occurrence in breast cancer?
5. How did the study account for inter-observer variability among the radiologists assessing the lesion conspicuity?
6. What measures were taken to minimize bias in the retrospective collection of data?
7. justify the use of univariate and multivariate analyses in their study? Could other statistical methods have been more appropriate?
8. provide more details on the statistical significance of the histological subtypes and their impact on lesion conspicuity?
9. What are the clinical implications of the findings for radiologists using CEM in routine practice?
10. What are the potential biases in the study due to its retrospective design, and how might these have been mitigated?
Author Response
- How was the sample size of 552 patients determined? Was a power analysis conducted to ensure adequate statistical power?
Dear reviewer, thank you for your question. A priori sample size was not determined. However, we acknowledge the importance of a power analysis to confirm adequate power was obtained to support the conclusions on the study. Post hoc power analysis was thus performed for the variables that were found to be statistically significant using the G*Power software v 3.1.9.4. The power analysis performed for those variables that showed significant difference yielded a 99% power for “Enhancement type on CEM” and “Dimensions”, 71% for “Ki67 index”, 83% for “Molecular subtype”, and 97% for “Histology”. We have added this information to the manuscript and discussed the result obtained.
Lines 169-170: “In addition, post hoc power analysis was performed for the variables that were found to be statistically significant using the G*Power software v 3.1.9.4.”
Lines 203-206: The power analysis performed for those variables that showed significant difference yielded a 99% power for “Enhancement type on CEM” and “Dimensions”, 71% for “Ki67 index”, 83% for “Molecular subtype”, and 97% for “Histology”.
Lines 314-317: Finally, the variable “Ki67 index” showed a power slightly below 80% suggesting that caution should be used when interpreting the results obtained and that further studies involving larger sample sizes should be performed.
- Clarify how lesion conspicuity was quantitatively measured or was it solely based on qualitative assessment?
The assessment of lesion conspicuity was solely based on qualitative evaluation, as outlined in the BIRADS lexicon and its definition. We have also included this clarification in the Materials and Methods section to ensure transparency.
- What specific criteria did the radiologists use to categorize lesion conspicuity into low, moderate, and high levels?
Thank you for your question; conspicuity was defined as low if the enhancement of the lesion was equal to or slightly greater than BPE, while high if the enhancement was much greater than BPE, and moderate if the enhancement was in between low and high, according to it definition in ACR lexicon supplement. We added this clarification also in M&M to ensure transparency.
- Why multifocal or multicentric diseases were excluded, given their common occurrence in breast cancer?
Multifocal or multicentric diseases were excluded due to the complexity they introduce in the analysis of lesion conspicuity. Since this is a technical paper on factors influencing the LC we thought that analysis could be more reliable when focusing on unifocal lesions. Multifocal and multicentric tumors are often heterogeneous, which complicates the assessment of conspicuity. To ensure a more coherent evaluation, these types of lesions were excluded from the study. This clarification has been added to the text, specifically at the end of the patient population section, to emphasize that the focus of the study was on achieving a more accurate analysis of the LC parameter.
- How did the study account for inter-observer variability among the radiologists assessing the lesion conspicuity?
In cases where there was no agreement among the radiologists regarding lesion conspicuity, the final decision was made through consensus. This approach has been added to the text for clarity.
- What measures were taken to minimize bias in the retrospective collection of data?
To minimize bias in the retrospective collection of data, contrast-enhanced mammography (CEM) was routinely performed on all patients in our unit, regardless of breast density, histotype, or tumor size. CEM is not used preferentially in dense breasts; rather, it is routinely conducted before surgery in our center. This has been clarified in the text to ensure transparency and avoid any misunderstanding.
- Justify the use of univariate and multivariate analyses in their study? Could other statistical methods have been more appropriate?
Thank you for your question. Univariate analyses were used to pre-screen the independent variables to include in the multivariate analysis. Alternatively, variables could be selected using a stepwise regression. However, stepwise regression has some disadvantages. The principal is that it depends on a computer algorithm to select the variables, without considering their relevance to the theoretical framework guiding the analysis. In addition, in this study we were interested in identifying which variables worthed further investigation but also in understanding the distribution of individual variables among the three lesion conspicuity groups and univariate analyses were deemed as useful to provide statistical summaries and visual interpretations of single variables.
- Provide more details on the statistical significance of the histological subtypes and their impact on lesion conspicuity?
The statistical analysis revealed that HER2+ tumors were more likely to exhibit high lesion conspicuity (LC). In contrast, Luminal A tumors were more commonly observed in the low LC group. We have clarified and expanded on these findings in the discussion section. Additionally, we noted that lobular carcinoma tends to distribute more uniformly across various LC degree groups, whereas mixed variants are more frequently seen in the low LC category. These points have been added to the discussion for a more comprehensive explanation.
- What are the clinical implications of the findings for radiologists using CEM in routine practice?
As mentioned in the discussion, the correlation between high conspicuity and aggressive molecular subtypes, although not an independent predictor, suggests CEM could help stratify patients by risk. High conspicuity may prompt more invasive diagnostic procedures, but while it’s often linked to malignancy, it’s not exclusive to malignant lesions. Benign lesions can also show high LC due to factors like size and vascularity, so high conspicuity should prompt further evaluation but not be seen as definitive for malignancy. Additionally, faintly enhancing low LC lesions should not be dismissed as benign without thorough assessment, as some cancers may present with subtle enhancement due to low vascularity or smaller size. Both highly conspicuous and faint lesions need careful evaluation, as relying solely on conspicuity could lead to missed diagnoses. A cautious approach to low-conspicuity lesions is recommended in pre-surgical planning.
- What are the potential biases in the study due to its retrospective design, and how might these have been mitigated?
Thank you for your question. The retrospective design of the study introduces potential biases, such as selection bias, as patients included may not represent the broader population due to the specific criteria used for inclusion. Additionally, information bias could arise from relying on existing records, which may have incomplete or inconsistent data. To mitigate these biases, all patients underwent routine CEM in our unit as locoregional presurgical staging, regardless of breast density or lesion characteristics, reducing selection bias. Moreover, we cross-verified key clinical and radiological data to minimize information bias. While retrospective studies inherently have limitations, these measures aimed to improve the reliability of the findings.
Reviewer 2 Report
Comments and Suggestions for Authors
The authors reported valuable data regarding lesion conspicuity on contrast-enhanced mammography. Overall, the article is well written and organized. However, the study is largely descriptive and conclusion is unclear. It is guaranteed that mass-forming lesions and advanced breast cancer are well enhanced on mammography because of neo-vascularization.
One of the major indications of contrast-enhanced mammography is to better characterize inconclusive lesion on the conventional mammography. Because there is no comparison between benign breast disease and malignancy in this study, we could not find clinical utility of contrast-enhanced mammography. However, these limitations of the study are well described.
Major comment:
1> This study was performed in the setting of breast cancer diagnosis. i.e. all the patients had been already diagnosed with breast cancer before contrast-enhanced mammography. Is there any added value of contrast-enhanced mammography for the patients already diagnosed with breast cancer? Did contrast-enhanced mammography identify additional occult lesion not identified on the conventional imaging?
Minor suggestions:
1> In Table 1, please use HER2, rather than Her2.
2> In Table 1, does “ductolobular invasive carcinoma” mean mixed ductal and lobular carcinoma?
3> In line 316, please finish the sentence after “Fine modulo”.
4> In reference 7, please write the title of the reference.
Author Response
One of the major indications of contrast-enhanced mammography is to better characterize inconclusive lesion on the conventional mammography. Because there is no comparison between benign breast disease and malignancy in this study, we could not find clinical utility of contrast-enhanced mammography. However, these limitations of the study are well described.
Thank you for your comment and for highlighting an important point. At this stage, the primary aim of our study was to evaluate lesion conspicuity (LC) in malignant lesions to identify associated factors. However, we agree that future research exploring whether LC could serve as a discriminative marker between benign and malignant lesions would indeed be very valuable.
- This study was performed in the setting of breast cancer diagnosis. i.e. all the patients had been already diagnosed with breast cancer before contrast-enhanced mammography. Is there any added value of contrast-enhanced mammography for the patients already diagnosed with breast cancer? Did contrast-enhanced mammography identify additional occult lesion not identified on the conventional imaging?
Thank you for your thoughtful questions. The primary aim of our study was not to assess the added value of contrast-enhanced mammography (CEM) in detecting additional occult lesions or to evaluate its role in patients already diagnosed with breast cancer. However, we acknowledge the importance of this aspect, and we have included a discussion in the introduction about the established role of CEM in pre-surgical planning.
- In Table 1, please use HER2, rather than Her2.
Thank you for your comment, we hace done the requested changes into text and tables.
- In Table 1, does “ductolobular invasive carcinoma” mean mixed ductal and lobular carcinoma?
Yes, we have changed the name according to your suggestion.
- In line 316, please finish the sentence after “Fine modulo”.
We apologize for the oversight in line 316 where the sentence after “Fine modulo” was left incomplete. It was an unintentional error, and we have now corrected it
- In reference 7, please write the title of the reference.
Thank you, we have corrected the error.
Round 2
Reviewer 1 Report
Comments and Suggestions for Authors
Accept in present form